# The M13 Phage Assembly Machine Has a Membrane-Spanning Oligomeric Ring Structure

**DOI:** 10.3390/v14061163

**Published:** 2022-05-27

**Authors:** Maximilian Haase, Lutz Tessmer, Lilian Köhnlechner, Andreas Kuhn

**Affiliations:** Institute of Biology, University of Hohenheim, 70599 Stuttgart, Germany; maximilian.haase@uni-hohenheim.de (M.H.); lutz.tessmer@uni-hohenheim.de (L.T.); lilikoehnlechner@gmx.de (L.K.)

**Keywords:** bacteriophage M13, membrane protein, affinity chromatography, circular dichroism, phage assembly machine

## Abstract

Bacteriophage M13 assembles its progeny particles in the inner membrane of the host. The major component of the assembly machine is G1p and together with G11p it generates an oligomeric structure with a pore-like inner cavity and an ATP hydrolysing domain. This allows the formation of the phage filament, which assembles multiple copies of the membrane-inserted major coat protein G8p around the extruding single-stranded circular DNA. The phage filament then passes through the G4p secretin that is localized in the outer membrane. Presumably, the inner membrane G1p/G11p and the outer G4p form a common complex. To unravel the structural details of the M13 assembly machine, we purified G1p from infected *E. coli* cells. The protein was overproduced together with G11p and solubilized from the membrane as a multimeric complex with a size of about 320 kDa. The complex revealed a pore-like structure with an outer diameter of about 12 nm, matching the dimensions of the outer membrane G4p secretin. The function of the M13 assembly machine for phage generation and secretion is discussed.

## 1. Introduction

Bacteriophage M13 contains only nine genes encoding 11 different proteins that control its propagation. Two of these proteins—namely G10p, the *gene 10* protein, and G11p—derive from internal start codons in *gene 2* and *gene 1*, respectively. Amazingly, the expression of these 11 proteins suffices to convert the *Escherichia coli* host cell into an extremely efficient phage production machinery [1,2]. How does the phage assemble in the inner membrane?

Initially, eight of the phage-encoded proteins are inserted as transmembrane proteins. G4p is localized in the outer membrane, whereas five of the others are designated as coat proteins for their assembly into nascent phage particles. G1p and G11p are required to catalyse the phage assembly in the inner membrane. The M13 G1p is synthesized as a single-spanning membrane protein of 348 residues where the N-terminal 253 residues are localized to the cytoplasm (Figure 1A). The N-terminal domain contains an ATP binding pocket that has an essential function in the assembly process of the phage [3]. Point mutations in the Walker A or B motifs lead to a severe inhibition of phage production, suggesting that the assembly and secretion of the M13 phage requires the hydrolysis of ATP by the M13 assembly machine. In addition to G1p, a shorter version of the protein is expressed from an internal start codon at position M241, named G11p [4], which lacks the ATP binding region but contains the transmembrane anchor region and the C-terminal periplasmic tail. It is assumed that G11p is part of the M13 assembly machine but its distinct role is still elusive.

The secretion of filamentous phage is linked to the assembly of the progeny particles in the inner membrane. During this process the extruding single-stranded DNA from the cytoplasm is coated by the proteins. First, the transmembrane G7p and G9p interact with the M13 assembly machine that has the single-stranded DNA bound at its morphogenetic signal [5]. The secretion of the progeny particle starts when multiple copies of the membrane-inserted major coat protein, G8p, are laterally fed into the M13 assembly machine [1,6]. All 2750 copies of G8p coat the DNA strand in a shingle-like helical arrangement and the growing particle leaves the inner membrane and moves across the outer membrane by extrusion through the inner cavity of the G4p secretin [7,8]. At the end of the secretion process the complex binds the transmembrane G3p and G6p to complete the formation of the particle. The latter two proteins constitute the phage base structure required for host adsorption in the next infection round.

Here, we employed a biochemical approach to purify and reconstitute the M13 assembly machine into pure lipid bilayers. This made it possible to study the structural and biochemical details of how this fascinating machine functions to catalyse the production of newly assembled phage particles. A model is presented to illustrate the individual steps involving the G1p/G11p complex during phage assembly and secretion.

## 2. Materials and Methods

### 2.1. Bacterial Strains, Phage and Plasmids

*E. coli* M15F^+^ was used for complementation and expression. This expression strain originates from *E. coli* M15 (Qiagen, Hilden, Germany) and contains an F-plasmid [3]. *Gene 1* of M13 was subcloned into pQE60 (Qiagen, Hilden, Germany) with an additional N-terminal 10xhis tag (MGH_10_SSGHIEGRHMLE), termed pQE60-HisG1/11, or with a C-terminal 9xhis tag (GSGENLYFQGGGSH_9_), termed pQE60-G1/11His. Bacteriophage M13 and the amber mutant in *gene 1* (M13am1) were from our collection. *E. coli* K37 was used as suppressor strain.

### 2.2. Purification of G1p/G11p

For the expression of G1p/G11p, *E. coli* M15F^+^ was transformed with pQE60-HisG1/11 and grown overnight in the presence of 200 µg/mL ampicillin at 37 °C. One litre of culture was inoculated (1:50) and grown at 37 °C under shaking (130 rpm) to a density of 2 × 10^8^ cells per mL. Then, the temperature of the culture was lowered to 15 °C, and induction with 0.05 mM IPTG and infection of M13am1 at an MOI of 5 was performed, followed by the continuation of the culture for 1 h at 15 °C. The cells were harvested at 6000× *g* for 15 min and resuspended in PBS buffer pH 7.5 (137 mM NaCl, 2.7 mM KCl, 10 mM NaPO_4_, 1.8 mM KH_2_PO_4_) containing 10% glycerol and 1 mM PMSF. The cells were lysed for four cycles in a French press (AIC, Haverhill MA, USA) at 500 psi. The membrane fraction was collected by centrifugation at 150,000× *g* at 4 °C for 1 h and resuspended in PBS buffer. For the solubilization 2% lauroyl sarcosine was added and incubated at 4 °C overnight. The solubilized G1p/G11p complex was then sedimented with 150,000× *g* at 4 °C for 1 h, followed by nickel-NTA affinity chromatography. Elution of the protein complex was initiated by 500 mM imidazole in a buffer containing 20 mM Tris-HCl pH 7.5, 300 mM NaCl, 10% glycerol and 0.6% lauroyl sarcosine. For further purification, size exclusion chromatography was performed with a Superdex 200 Increase 30/10 (Cytiva, Marlborough, MA, USA).

### 2.3. Circular Dichroism

The secondary structure of the purified G1p/G11p was determined using a Jasco 715-CD spectrometer (Hajioji, Japan). For these measurements, the buffer of purified protein was exchanged with 10 mM sodium phosphate buffer pH 7.5. Structural analysis was performed at 25 °C with 10 measurements each of three different samples. Thermal denaturation of the sample was performed in 0.5 °C steps. Based on the resulting CD spectrum, the approximate content of α-helices and β-sheets was calculated using the K2D2 algorithm [9].

### 2.4. Reconstitution into Liposomes

Multilaminar POPC vesicles were used for reconstitution of the G1p/G11p complex into unilaminar liposomes. At a ratio of 1:10,000, 1 µM G1p/G11p was incubated with the lipid (buffer: 10 mM Tris-HCl pH 7.5 and 150 mM NaCl) for 10 min at 37 °C and extruded through a 0.4 µm nitrocellulose membrane. The resulting proteoliposomes had a diameter of~400 nm. The remaining detergent was removed by washing with bio-beads (Bio-Rad Laboratories, Hercules, CA, USA) twice, each for 1 h at 4 °C. To confirm the successful reconstitution, the proteoliposomes were centrifuged at 100,000× *g* for 15 min at 4 °C and the two fractions, supernatant and pellet, were analysed with SDS-PAGE. The diameter of the proteoliposomes was determined with dynamic light scattering (Avid-Nano, High Wicombe, UK).

### 2.5. Purification of G5p-ssDNA

Expression of G5p was performed with *E. coli* BL21 bearing the plasmid pMS119HE-G5p with an N-terminal 10xhis tag. An overnight culture was used 1:50 to inoculate 1 L LB medium, shaken to a density of 2 × 10^8^ at 37 °C and induced with 0.5 mM IPTG. The culture was continued for 2 h at 37 °C and the cells were pelleted and resuspended in 20 mM Tris-HCl, pH 7.5 and 150 mM NaCl. The cells were opened with a French press at 1.3 kBar. G5p was affinity-purified using a nickel-NTA column. 

M13 ssDNA was purified from M13 phage with a ssDNA prep kit (Qiagen, Hilden, Germany). For the formation of the complex between ssDNA and G5p, both were incubated for 15 min at RT. Binding experiments of G5p-ssDNA with proteoliposomes were performed with 200 ng ssDNA and 0.6 µg G5p.

### 2.6. AFM and Electron Microscopy

For AFM images, *E. coli* M15F^+^ pQE60-G1/11 was infected with M13am1 phage and collected by centrifugation after 30 min. The bacterial pellet was washed with ddH_2_O twice and 5 µL were applied to mica. The mica was cleaned after 5 min by immersion in a beaker with water and then dried. Images were acquired with the AFM Nanoscope III (Digital Instruments) using the tapping mode and deflection image with the parameters 12 Hz and 512 lines.

TEM images of the G1p/G11p complex were obtained by negative staining with 2% uranyl acetate. On a 400-mesh grid, 5 µL of the sample (c = 10 µg/mL) was applied and incubated for 2 min. The grid was washed three times with ddH_2_O and stained for 5 min on a drop with uranyl acetate. Images were acquired using a Leo912AB electron microscope (Leica/Zeiss) with a voltage of 100 kV and a magnification of 31,500×.

## 3. Results

### 3.1. Expression of G1p and G11p

Since G1p and G11p are expressed from the same gene (Figure 1A) we expected the plasmid encoded proteins to be equally overproduced. The expression from the plasmid is under *lac* control and can be induced with IPTG. Indeed, immunoprecipitations show that both proteins were expressed after induction with 0.5 mM IPTG for 1 h (Figure 2A). To purify G1p, a his tag sequence was added either to the N-terminal or the C-terminal end of the gene, respectively. Complementation by the plasmid-encoded proteins was tested on *E. coli* M15F^+^ lawns bearing the respective plasmid in the presence of 0.025 mM IPTG or in the absence of IPTG as a control (Figure 2B). Serial dilutions of M13am1 phage were spotted onto the top agar containing the bacteria and incubated overnight. The plasmid-encoded G1p/G11p fully complemented the amber mutation showing an identical number of plaques as the suppressor lawn K37. The addition of the tag still allowed complementation but reduced the efficiency of plaque formation by the M13am1 phage.

The complementation by the plasmid-encoded G1p/G11p allowed us to visualize the progeny production of M13am1 by AFM (Figure 1B). An M15F^+^ culture bearing the plasmid pQE60 encoding G1p/G11p was infected with M13am1 for 30 min. An aliquot of the culture was placed onto mica and processed to visualize the cells. Numerous phage progeny were detected in their secretion processes, corroborating an efficient complementation.

### 3.2. Purification of the M13 Assembly Machine

The 10xhis tag at the N-terminus of G1p was added to purify a high-molecular-weight complex by affinity chromatography. *E. coli* M15F+ cells bearing a plasmid encoding HisG1p and G11p were grown to the exponential phase. Then, the temperature of the culture was lowered to 15 °C, induced with 0.05 mM IPTG and infected with M13am1 at an MOI of 5. The culture was continued for 1 h at 15 °C under aeration and harvested by centrifugation. The cells were lysed by French press and after the removal of large debris the membrane fraction was collected and solubilised by incubation with 2% lauroyl sarcosine overnight. After removal of the unextracted membranes the supernatant was mixed with Ni-NTA beads in the presence of 10 mM imidazole and 10% glycerol for 1.5 h at 4 °C. The protein eluted with 100 mM imidazole and the fractions were analysed by SDS-PAGE (Figure 3A). The two proteins G1p and G11p eluted in the same fractions. Since only G1p has a his tag, the presence of G11p in the same fraction shows that both proteins are part of the same complex.

Similar results were obtained with the C-terminally tagged G1p/G11p (Figure 3B). As shown by size exclusion chromatography, both versions formed oligomeric complexes that eluted at 9.8 mL (Figure 3C) and 9.7 mL (Figure 3D), respectively, which corresponds to a molecular weight of about 320 kDa.

### 3.3. Circular Dichroism Shows a Mainly α-Helical Structure 

To investigate the secondary structure and the conformational stability of G1p/G11p, the purified complex was analysed with circular dichroism (CD). The protein showed a high α-helical content of 57.5% and a β-sheet content of 6.5% (Figure 4A). The thermal stability was tested at 220 nm and showed that the unfolding of the α-helical content began above 50 °C and continued until 80 °C (Figure 4B). This resulted in a transition point (Tm) at 65 °C. The unfolded protein did not show any refolding when the temperature was again lowered (Figure 4C).

### 3.4. Reconstitution of G1p/G11p into Proteoliposomes

To reconstitute the purified complex into a lipid bilayer, the protein was mixed with phosphatidylcholine (PC) and extruded to generate proteoliposomes. To test whether the protein had integrated into the lipid bilayer, the proteoliposomes were sedimented by centrifugation and analysed with SDS-PAGE (Figure 5). Most of the G1p complex was found in the pellet fraction, suggesting that the reconstitution into the proteoliposomes was efficient.

### 3.5. Purification and DNA Binding of the M13 G5p

The DNA binding protein G5p with a C-terminal 10xhis tag was purified using nickel affinity chromatography (Figure 6A,B). Its detection was verified with Coomassie staining and a Western blot, with an antibody against the His tag showing a distinct band at 10 kDa. The purified protein was added to purified M13 ssDNA and its binding to the DNA was analysed with the electrophoretic mobility shift assay (EMSA). Increasing amounts of the G5p showed a shift of the DNA band (Figure 6C), corroborating the efficient binding to the ssDNA.

### 3.6. Binding of the ssDNA to the G1p/G11p Complex

Since the replicated progeny DNA binds G5p prior to its packaging, we were interested whether it could bind to the purified and reconstituted G1p/G11p complexes. Therefore, the single-stranded DNA was extracted from the M13 phage and incubated with purified G5p. The purified G5p without DNA did not bind to liposomes and was found in the supernatant after a high-speed centrifugation (Figure 7, lane 1). Furthermore, it did not bind to proteoliposomes containing G1p/G11p and remained in the supernatant (lane 3). However, when we used the G5p-ssDNA complex, the protein was found in the pellet fraction (lane 8), suggesting the efficient binding of the G5p-ssDNA complex to the proteoliposomes, whereas no binding was observed to empty liposomes (lane 5).

### 3.7. Electron Microscopy of G1p/G11p Complexes

The purified G1p/G11p complex was inspected with electron microscopy. To do this, the peak sample of the complex after gel filtration (Figure 3C) was applied to a 400-mesh grid and negatively stained with 2% uranyl acetate. Uniform particles with a diameter of about 12 nm were detected, and some of the particles appeared hexameric and had an inner pore structure (Figure 8). Notably, some of the particles showed one to three prominent white dots, which might have indicated an extended domain out of the pore structure.

## 4. Discussion

The filamentous phages (M13, fd, f1) are assembled in the inner bacterial membrane with the participation of a multimeric assembly machinery. This G1p/G11p complex binds the newly replicated ssDNA at its cytoplasmic side and binds transmembrane coat proteins entering the complex from its lateral side. This leads to the coating of the ssDNA as it is funnelled through the pore of the complex. In essence, the assembly is a sequentially ordered process within this machinery. How this machinery selects and binds the transmembrane coat proteins and allows the lateral entrance of the phage coat proteins to contact the DNA is presently unknown. Since the assembled coat subunits bind in a helical order on the phage, the formation of this helix might turn the nascent phage in the pore structure and power the movement and secretion out of the cell [1]. In addition, the hydrolysis of ATP that is catalysed by the N-terminal domain of G1p possibly moves the ssDNA through the secretion complex. A structural analysis at the atomic level could shed light on this interesting process in the future.

The purification and size exclusion chromatography of the G1p/G11p complex showed an apparent molecular weight of about 320 kDa. This could theoretically correspond to six copies of G1p and six copies of G11p, suggesting a hexameric composition (Figure 3). An oligomeric appearance was also supported by an electron microscopic analysis of the purified complex (Figure 8). We show here that the G1p/G11p complex appears as a ring structure, with a central cavity of about 5 nm in diameter that could host a phage filament during its assembly since a phage filament has a diameter of 6 nm [10]. The particles also showed white dots, which might have indicated protrusions, possibly the ATPase domains of G1p that are located in the cytoplasm [3]. The outer diameter of the particles was determined to be about 12 nm, which fit with the dimensions of its counterpart in the outer membrane, the G4p secretin [7]. The secretin has an outer diameter of 11 nm and is composed of fifteen subunits, each spanning the outer membrane and with 4 β-strands resulting in a porin-like structure containing 60 β-strands.

It has been shown very early on with genetic methods by Marjorie Russel that G4p and G1p interact in the periplasm [11]. Later, this was corroborated biochemically by showing that the N-terminal domain of G4p can be cross-linked with G1p [12]. Consequently, the extruding phage filaments can pass from their origin in the inner membrane through the outer membrane without interruption. However, in our isolation procedure we did not find G4p in complex with G1p/G11p (Figure 3A), suggesting that the interaction between the G1p/G11p machinery and the G4p secretin is weak and lost during the isolation. If the G1p/G11p complex has a six-fold symmetry, the five-fold symmetry of the secretin would create a mismatch. Symmetry mismatches are not uncommon in phage systems and we find such mismatches in phage packaging systems [13] and head-portal connections [14]. If so, such a mismatch would explain why G4p secretin binds the G1p/G11p complex with a rather low affinity.

The newly replicated single-stranded DNA is first present in the cytoplasm as a nucleoprotein complex coated with G5p [15,16]. During the assembly process of the phage, the G5p bound on the DNA is replaced by the coat proteins. For the M13 DNA, it has been shown that a specific packaging signal is present and required [17]. Presumably, this structure enters the assembly machine first and initiates the phage assembly process. Indeed, we observed efficient binding of the G5p-ssDNA to the reconstituted G1p/G11p complex in proteoliposomes (Figure 7). In the next step of the M13 phage assembly pathway the two minor coat proteins G7p and G9p enter the assembly machine laterally and form a cap structure (Figure 9A). G7p and G9p are small single-spanning membrane proteins of 33 and 32 amino acid residues, respectively. G7p has been shown to insert into the inner membrane of *E. coli* without the assistance of an insertase or translocase [18], whereas G9p requires YidC [19]. Both proteins then bind with their C-terminal positively charged region to the DNA, most likely at the cytoplasmic side in the pore region of the assembly machinery (Figure 9B). In fact, it has been shown that both proteins can be used for phage display after extending their N-termini [19,20] in contrast to their C-termini, which cannot be altered. These observations corroborate that the C-terminal parts of G7p and G9p are dedicated to DNA binding. In the next step, the nascent phage is coated with approximately 2750 copies of the major coat protein G8p (Figure 9C). Similarly to G7p and G9p, the C-terminal part of G8p is positively charged and binds to the phage DNA, and the positively charged residues play an important role in DNA binding [21]. During the further assembly process of the phage, G5p binds on the DNA and is gradualy replaced by the major coat proteins (Figure 9D). Finally, five copies each of G3p and G6p, which also enter the assembly machine laterally, terminate the assembly process. The phage particles are extruded through the central pore of the G4p secretin complex in the outer membrane and released into the external medium.

## 5. Conclusions

The phage assembly machine of M13 is localized in the inner membrane as an oligomeric ring structure composed of G1p and G11p. Here, we describe the isolation and membrane reconstitution of the complex that is competent to bind M13 single-stranded phage DNA together with its binding protein G5p. An assembly model is proposed where the complex then binds the minor coat proteins initiating the secretion process. The subsequent assembly steps then involve the binding of the 2750 copies of the major coat proteins and the two terminating proteins at the base of the phage particle.

## Figures and Tables

**Figure 1 viruses-14-01163-f001:**
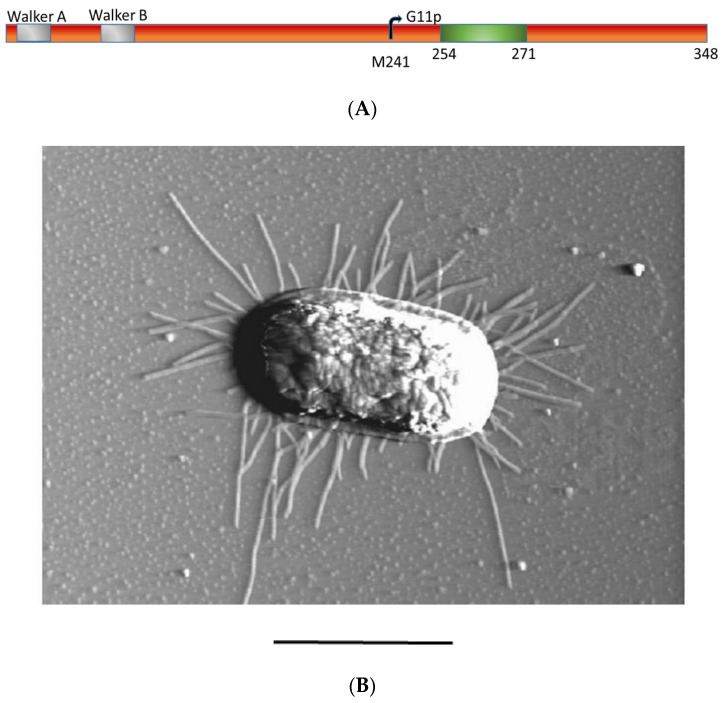
The secretion of M13 phage by *E. coli* is mediated by the M13 assembly machine. (**A**) Schematic representation of the G1p protein regions with the indicated Walker box sequences (grey, residues 8 to 15 and 84 to 89), the internal start codon M241 for G11p and the membrane anchor segment (green, 254 to 271) and the periplasmic domain (residues 272 to 348). (**B**) An M13-secreting *E. coli* cell 10 min post-infection was visualized by atomic force microscopy (AFM). The bar represents 1 µm.

**Figure 2 viruses-14-01163-f002:**
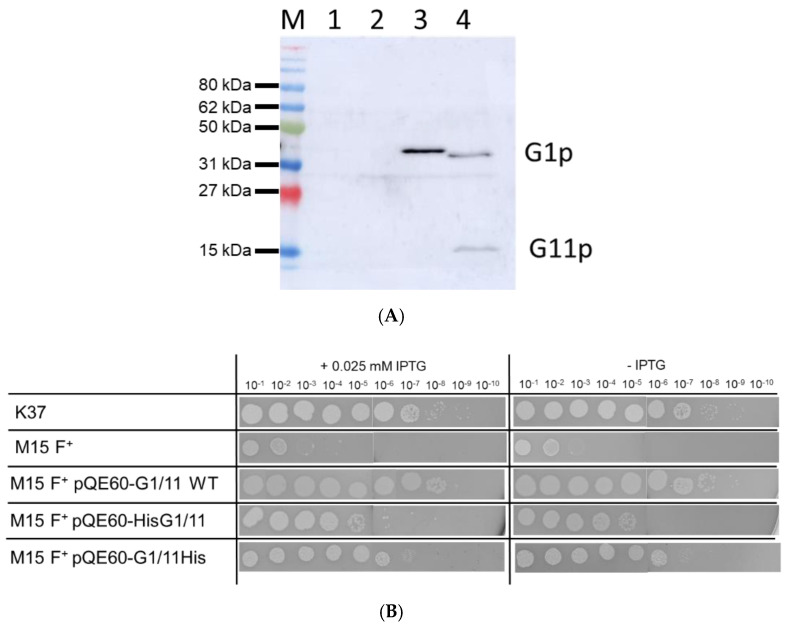
Expression and functionality of plasmid-encoded G1p. (**A**) The expression of the plasmid-encoded G1p was analysed in M15F+ cells by Western blot. Lane 1: empty plasmid, lane 2: G1p/G11p, lane 3: HisG1p/G11p and lane 4: G1pHis/G11pHis were expressed from the respective plasmids after induction with 0.5 mM IPTG for 1 h and processed for immunoprecipitation. (**B**) The plasmids encoding G1p and G11p were transformed into M15F+ cells and analysed for complementation of M13am1 phage propagation, either with 0.025 IPTG (left column) or no IPTG (right column). For a control, *E. coli* K37 (sup^+^) was included in the analysis.

**Figure 3 viruses-14-01163-f003:**
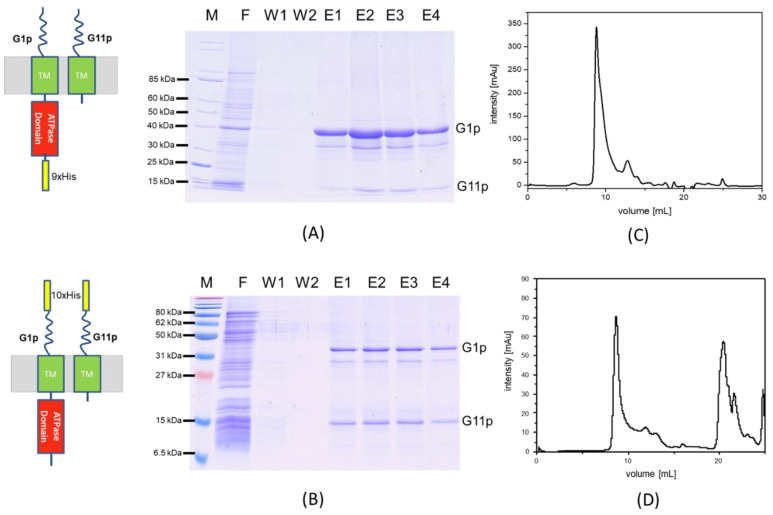
Purification of G1p/G11p by Ni^2+^-affinity chromatography. The HisG1p and G11p elute in the same fractions (**A**), similarly to the C-his construct (**B**) as the Coomassie stained gels show. Size exclusion chromatography with a Superdex 200 10/30 column was used to further purify the N-his complex (**C**) and the C-his complex (**D**). Both purification profiles show a single peak at an elution volume of about 10 mL, corresponding to a molecular weight of 320 kDa. M refers to Marker, F to flow through, W to wash and E to elution fraction.

**Figure 4 viruses-14-01163-f004:**
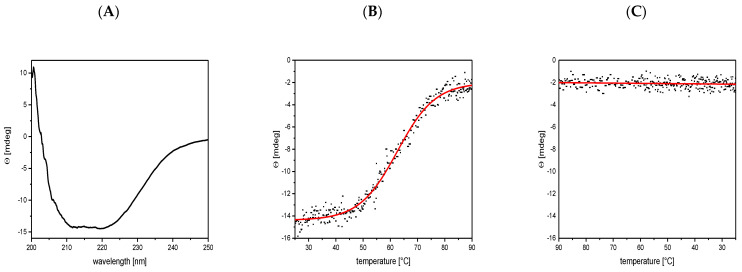
The G1p/G11p contains a high α-helical content and shows thermal stability with a transition point of Tm = 65 °C. (**A**) The purified G1p was analysed for its secondary structure with circular dichroism (CD). (**B**) The unfolding of the α-helical content was followed at 220 nm with increasing temperature, showing a transition point at 65 °C. (**C**) Refolding of the unfolded protein was measured at 220 nm from 90 °C to 25 °C.

**Figure 5 viruses-14-01163-f005:**
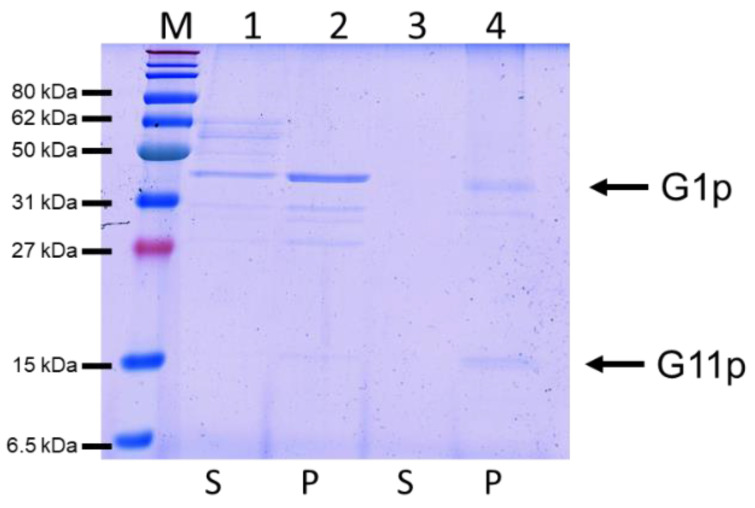
Reconstitution of HisG1p/G11p (lanes 1, 2) and G1p/G11pHis (lanes 3, 4) into PC liposomes to generate proteoliposomes. After sedimentation of the proteoliposomes, G1p and G11p were mainly found in the pellet (P) fraction and not in the supernatant (S).

**Figure 6 viruses-14-01163-f006:**
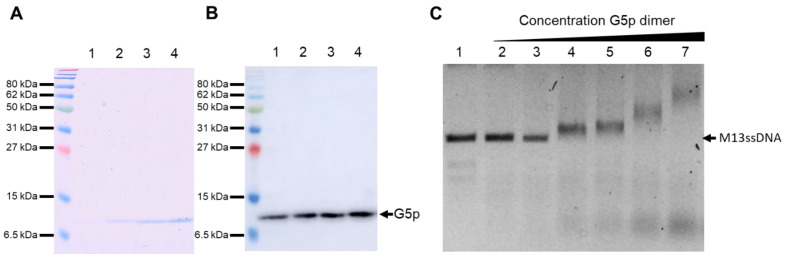
Affinity purification of the DNA-binding protein G5p. The elution fractions (lanes 1 to 4) were visualized with Coomassie staining (**A**) and on a Western blot with a his tag antibody (**B**). The purified G5p was incubated with purified M13-ssDNA and analysed with EMSA (**C**). Lanes 1 to 8 contained 200 ng of M13-ssDNA. Increasing amounts of G5p were added; in lane 2: 15 ng, lane 3: 80 ng, lane 4: 130 ng, lane 5: 160 ng, lane 6: 480 ng and lane 7: 800 ng.

**Figure 7 viruses-14-01163-f007:**
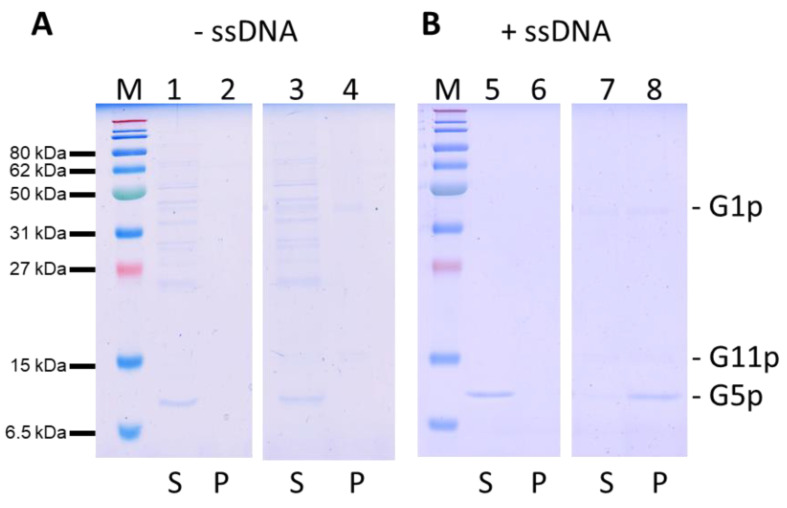
Binding of G5p-ssDNA to proteoliposomes containing G1p/G11pHis. (**A**) Purified G5p was added to liposomes (lanes 1, 2) or to proteoliposomes containing G1p/G11p (lanes 3, 4). (**B**) G5p-ssDNA complexes were added to liposomes (lanes 5, 6) or proteoliposomes (lanes 7, 8). After the vesicles were spun down, the supernatant (S, odd lanes) and pellet (P, even lanes) were analysed for their protein content with PAGE and Coomassie staining.

**Figure 8 viruses-14-01163-f008:**
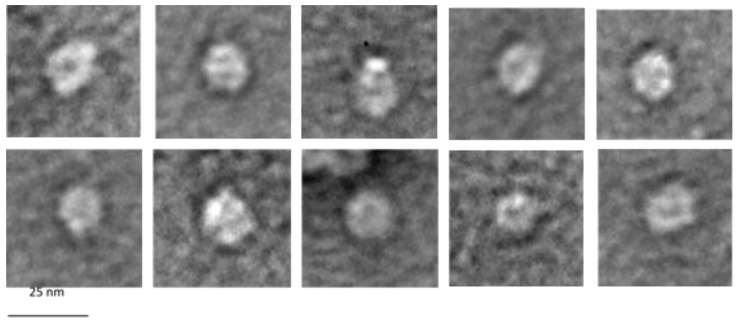
Electron micrographs of G1p/G11p complexes. The purified G1p/G11p was applied to a 400-mesh grid, negatively stained with UAc and analysed by electron microscopy. Note the white dots that appear in some of the complexes.

**Figure 9 viruses-14-01163-f009:**
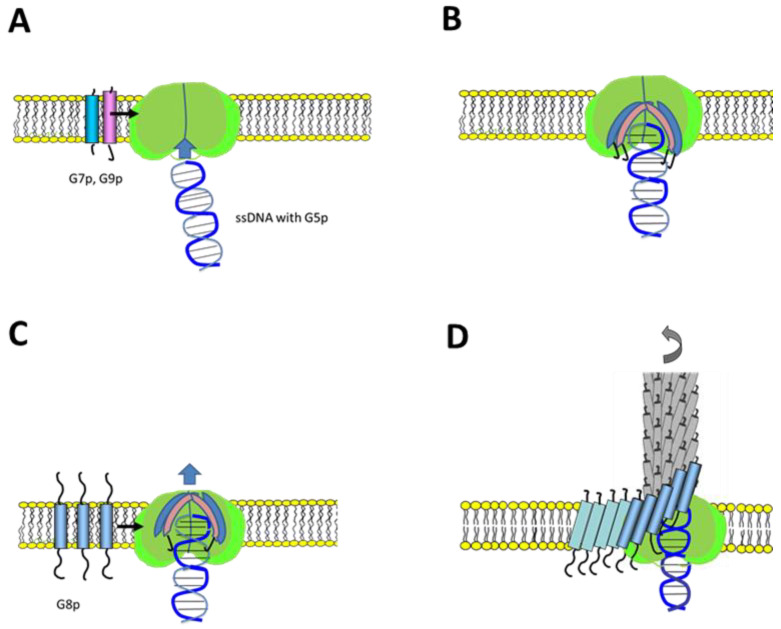
Model of M13 phage assembly steps. (**A**) Binding of the single-stranded DNA to the G1p/G11p complex. (**B**) Lateral entry of the transmembrane G7p and G9p proteins into the G1p/G11p complex (green). (**C**) Lateral entry of multiple G8p proteins into the assembly complex. (**D**) Movement of the nascent phage out of the pore structure.

## Data Availability

All data presented were done by following the MDPI Research Data rules.

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
