# Peer review of "The M13 Phage Assembly Machine Has a Membrane-Spanning Oligomeric Ring Structure"

_viruses, 2022, doi:10.3390/v14061163_

Round 1
Reviewer 1 Report
In this manuscript, the authors described how they purified and characterized the M13 phage assembly machine. They isolated the G1p/G11p complex and performed several preliminary characterizations. In general, I think the authors had a great vision but did not provide enough data to support several of their claims. Also, this manuscript contains quite a few sentences that are either irrelevant to the actual experimental work or disconnected from the narratives. The following points should be addressed to increase the quality of this paper.
- The title is quite general about the entire phage assembly machine, while the actual work is not. Please consider using a more specific title that better captures the findings of this work.
- Line 39-49: Please include a simple diagram visualizing the protein-protein interactions. The readers might find it challenging to follow the current pure-text description.
- 1B and Fig. 3B require more description in the Results section.
- 2A: Could the authors comment on the different MW of G1p observed in lanes 3 and 4?
- Line 135-140: Please comment on the role of IPTG induction in this experiment as it is not very self-evident from the image.
- Line 162-173: Most of these should be moved to the Methods section.
- 3A: Please explain what the lane labels mean in the caption.
- Section 3.4: Please describe the data presented in Fig.5
- 5: Please make sure to use the same MW marker labeling format as shown in previous figures.
- Line 254-264: Many sentences repeat what’s covered in the previous sections. Please consider removing them or moving them to the Introduction section.
- Line 269-271: It is unclear how the authors derive such details from the low-resolution TEM images (please cite Fig. 6). Could the authors cite more relevant references on using low-resolution TEM images to infer such geometrical details?
- Line 280-282: The data shown in Fig. 3A are insufficient to support the claim that G4p (or any other viral proteins) is not present. Please include data collected from assays with higher specificity, such as Western blotting.
- Line 287-300: These two paragraphs are not related to any experimental work presented and would only distract the readers. The authors should revise or remove them.
- Line 305-306: I could not find any data in this work that support such a claim about ssDNA binding. The last sentence also seems to be irrelevant to the experimental findings of this work. This section should be revised.
Author Response
In this manuscript, the authors described how they purified and characterized the M13 phage assembly machine. They isolated the G1p/G11p complex and performed several preliminary characterizations. In general, I think the authors had a great vision but did not provide enough data to support several of their claims. Also, this manuscript contains quite a few sentences that are either irrelevant to the actual experimental work or disconnected from the narratives. The following points should be addressed to increase the quality of this paper.
- The title is quite general about the entire phage assembly machine, while the actual work is not. Please consider using a more specific title that better captures the findings of this work.
Answer: We changed the title to: The M13 phage assembly machine is a membrane-spanning oligomeric ring structure.
- Line 39-49: Please include a simple diagram visualizing the protein-protein interactions. The readers might find it challenging to follow the current pure-text description.
Answer: We now added a new figure in the discussion section where this assembly process is discussed in more detail.
- 1B and Fig. 3B require more description in the Results section.
Answer: For each figure a paragraph is now added after line 148 and 177, respectively.
- 2A: Could the authors comment on the different MW of G1p observed in lanes 3 and 4?
Answer: The N-terminal His-tag has a slightly higher MW compared to the C-terminal tag. In addition, we suspect a conformational effect of the tags on the migration in the gel. We have added the precise sequence of the tags in line 81/82.
- Line 135-140: Please comment on the role of IPTG induction in this experiment as it is not very self-evident from the image.
Answer: The IPTG induces the expression of G1p/G11p from the plasmid. We have added a sentence mentioning this.
- Line 162-173: Most of these should be moved to the Methods section.
Answer: We have shortened the technical details in order to avoid repetitious mentioning.
- 3A: Please explain what the lane labels mean in the caption.
Answer: We have added this in the caption to Fig. 3
- Section 3.4: Please describe the data presented in Fig.5
Answer: We added this in more detail.
- 5: Please make sure to use the same MW marker labeling format as shown in previous figures.
Answer: We corrected this.
- Line 254-264: Many sentences repeat what’s covered in the previous sections. Please consider removing them or moving them to the Introduction section.
Answer: We have shortened this part at the beginning of the discussion.
- Line 269-271: It is unclear how the authors derive such details from the low-resolution TEM images (please cite Fig. 6). Could the authors cite more relevant references on using low-resolution TEM images to infer such geometrical details?
Answer: We have changed the sentence with the geometrical details.
- Line 280-282: The data shown in Fig. 3A are insufficient to support the claim that G4p (or any other viral proteins) is not present. Please include data collected from assays with higher specificity, such as Western blotting.
Answer: G4p is a 50 kDa protein and easy to see on a Coomassie gel. Fig. 3A does not show a band at 50 kDa. If the G4p complex would bind to the G1p/G11p complex it would show a strong band since the G4p secretin complex is a 15mer. We changed the respective sentence to “mainly lost during isolation” to point out that we do not address minute G4p amount that would only be detected by a Western.
- Line 287-300: These two paragraphs are not related to any experimental work presented and would only distract the readers. The authors should revise or remove them.
Answer: We have added a new figure and in the light of the proposed phage assembly model these two paragraphs are now better embedded into the discussion.
- Line 305-306: I could not find any data in this work that support such a claim about ssDNA binding. The last sentence also seems to be irrelevant to the experimental findings of this work. This section should be revised.
Answer: We have added new data (Figures 6 and 7) showing the G5p-ssDNA binds to the assembly machine. Accordingly, we have changed this part of the manuscript.
Reviewer 2 Report
The infection, assembly and propagation mechanism of bacteriophage M13 is a fascinating research topic to study on. This manuscript mainly described the overproduction, isolation and membrane reconstitution of the phage assembly machine, i.e., the multimeric complex of G1p and G11p with the size of ~320kDa. The secondary structure and conformational stability of G1p and G11p complex were determined by circular dichroism, and the structure was confirmed by electron microscopy.
This manuscript seems quite appealing at first, especially at the end of the introduction section, the author claimed that the reconstitution of the M13 assembly machine into pure lipid bilayers will “allow to study the structural and biochemical details of how the machine functions catalysing the production of newly assembled phage particles”. However, the study just stops at the verification of the reconstitution of G1p/G11p into proteoliposomes (section 3.4 L214-228). It is thus very confusing for readers to get the main purpose or achievement of this research. For me, what can be certain is that the author has successfully purified the G1p/G11p complex and did initial characterization, but that is not enough for a complete research. If the author focuses on studying the biochemical properties of the G1p/G11p complex or reconstituted proteoliposomes, then more detailed and deeper investigation is required. Therefore, I recommend the author either to reorganize this manuscript or get deeper investigations before organized the research results into a paper.
Specific comments:
- In the introduction section, there were so much words explaining the general process of phage assembly. To help the readers to clearly understand the process, a schematic diagram was suggested in this section. And for Abstract, the background introduction is too much, which account for ~60% of the whole abstract, the author should focus more on the main achievement or conclusion of this study.
- The discussion section is mainly used for detailed discussion and explanation of the experiments results obtained in section 3. However, most of the results in section 3 is not fully discussed.
- What is “S” standing for in Figure 5? This should be addressed in the caption of Figure 5. Besides, brief description should also be added to explain how can we demonstrate the proteoliposomes complex was successful prepared from Figure 5. And the purpose of preparing proteoliposomes complex also need to be explained. The molecular weight of proteoliposomes complex was between 31kDa and 50kDa as Figure 5 indicated, why only G1p protein was packaged in the liposomes, while G11p was not? And other characterization methods for proteoliposomes complex should also be added to demonstrated the successful reconstitution of proteoliposomes.
- The electron micrographs of G1p/G11p complexes of Figure 6 were too vague, and it is hard for readers to discriminate “hexameric” from such a vague image. Maybe a cryoelectron microscopy analysis will be better.
- The purification process of G1p/G11p complex was mentioned in the section of ‘Material and Methods’ (P 3, L 86), while the same context showed up in the 162 lines of paper 5. The repeated description is suggested to be removed.
- The “5 five copies” needed to be corrected as ‘five copies’ (P7, L258).
Author Response
The infection, assembly and propagation mechanism of bacteriophage M13 is a fascinating research topic to study on. This manuscript mainly described the overproduction, isolation and membrane reconstitution of the phage assembly machine, i.e., the multimeric complex of G1p and G11p with the size of ~320kDa. The secondary structure and conformational stability of G1p and G11p complex were determined by circular dichroism, and the structure was confirmed by electron microscopy.
This manuscript seems quite appealing at first, especially at the end of the introduction section, the author claimed that the reconstitution of the M13 assembly machine into pure lipid bilayers will “allow to study the structural and biochemical details of how the machine functions catalysing the production of newly assembled phage particles”. However, the study just stops at the verification of the reconstitution of G1p/G11p into proteoliposomes (section 3.4 L214-228). It is thus very confusing for readers to get the main purpose or achievement of this research. For me, what can be certain is that the author has successfully purified the G1p/G11p complex and did initial characterization, but that is not enough for a complete research. If the author focuses on studying the biochemical properties of the G1p/G11p complex or reconstituted proteoliposomes, then more detailed and deeper investigation is required. Therefore, I recommend the author either to reorganize this manuscript or get deeper investigations before organized the research results into a paper.
Answer: We have added new results (Figures 6 and 7) and think that this puts the scope of the paper into a more complete story. We thank the reviewer encouraging us to do this.
Specific comments:
- In the introduction section, there were so much words explaining the general process of phage assembly. To help the readers to clearly understand the process, a schematic diagram was suggested in this section. And for Abstract, the background introduction is too much, which account for ~60% of the whole abstract, the author should focus more on the main achievement or conclusion of this study.
Answer: We have added a schematic figure at the end of the paper to enhance the understanding of the process. We also changed the abstract
- The discussion section is mainly used for detailed discussion and explanation of the experiments results obtained in section 3. However, most of the results in section 3 is not fully discussed.
Answer: We considerably change the results sections and the discussion parts.
- What is “S” standing for in Figure 5? This should be addressed in the caption of Figure 5. Besides, brief description should also be added to explain how can we demonstrate the proteoliposomes complex was successful prepared from Figure 5. And the purpose of preparing proteoliposomes complex also need to be explained. The molecular weight of proteoliposomes complex was between 31kDa and 50kDa as Figure 5 indicated, why only G1p protein was packaged in the liposomes, while G11p was not? And other characterization methods for proteoliposomes complex should also be added to demonstrated the successful reconstitution of proteoliposomes.
Answer: We added the missing information into the caption of Fig. 5. The reconstituted complex in the liposomes was analysed after denaturation on an SDS-PAGE. We have repeated the experiment and replaced the figure which now shows G1p and G11p.
- The electron micrographs of G1p/G11p complexes of Figure 6 were too vague, and it is hard for readers to discriminate “hexameric” from such a vague image. Maybe a cryoelectron microscopy analysis will be better.
Answer: We fully agree with the reviewer, but this is a rather long-termed extension of the project.Still, we think that a basic low resolution structure of the complex showing a pore is interesting to have.
- The purification process of G1p/G11p complex was mentioned in the section of ‘Material and Methods’ (P 3, L 86), while the same context showed up in the 162 lines of paper 5. The repeated description is suggested to be removed.
Answer: We have changed the respective part (line 162).
- The “5 five copies” needed to be corrected as ‘five copies’ (P7, L258).
Answer: Corrected.
Round 2
Reviewer 1 Report
The authors have improved the clarity of the manuscript significantly. I think this paper is ready to be published.
Reviewer 2 Report
The manuscript is ready for publication.